# Characterization of a Preclinical In Vitro Model Derived from a SMARCA4-Mutated Sinonasal Teratocarcinosarcoma

**DOI:** 10.3390/cells13010081

**Published:** 2023-12-30

**Authors:** Sara Lucila Lorenzo-Guerra, Helena Codina-Martínez, Laura Suárez-Fernández, Virginia N. Cabal, Rocío García-Marín, Cristina Riobello, Blanca Vivanco, Verónica Blanco-Lorenzo, Paula Sánchez-Fernández, Fernando López, Jóse Luis Llorente, Mario A. Hermsen

**Affiliations:** 1Department of Head and Neck Cancer, Health Research Institute of the Principality of Asturias, 33011 Oviedo, Spain; saralgcv96@gmail.com (S.L.L.-G.); helenacm14@gmail.com (H.C.-M.); suarezflaura@gmail.com (L.S.-F.); vircabal@hotmail.com (V.N.C.); rociogm220879@hotmail.com (R.G.-M.); cristinarisu15@gmail.com (C.R.); 2Department of Pathology, Hospital Universitario Central de Asturias, 33011 Oviedo, Spain; vivancoblanca@gmail.com (B.V.);; 3Department of Otolaryngology, Hospital Universitario Central de Asturias, 33011 Oviedo, Spain; psfernan90@hotmail.com (P.S.-F.); flopez_1981@yahoo.es (F.L.); llorentejlx@gmail.com (J.L.L.)

**Keywords:** sinonasal cancer, teratocarcinosarcoma, in vitro model, exome sequencing, SMARCA4 mutation, preclinical drug testing

## Abstract

Sinonasal teratocarcinosarcoma (TCS) is a rare tumor that displays a variable histology with admixtures of epithelial, mesenchymal, neuroendocrine and germ cell elements. Facing a very poor prognosis, patients with TCS are in need of new options for treatment. Recently identified recurrent mutations in *SMARCA4* may serve as target for modern therapies with EZH1/2 and CDK4/6 inhibitors. Here, we present the first in vitro cell line TCS627, established from a previously untreated primary TCS originating in the ethmoid sinus with invasion into the brain. The cultured cells expressed immunohistochemical markers, indicating differentiation of epithelial, neuroepithelial, sarcomatous and teratomatous components. Whole-exome sequencing revealed 99 somatic mutations including *SMARCA4*, *ARID2*, *TET2*, *CDKN2A*, *WNT7A*, *NOTCH3* and *STAG2*, all present both in the primary tumor and in the cell line. Focusing on mutated *SMARCA4* as the therapeutic target, growth inhibition assays showed a strong response to the CDK4/6 inhibitor palbociclib, but much less to the EZH1/2 inhibitor valemetostat. In conclusion, cell line TCS627 carries both histologic and genetic features characteristic of TCS and is a valuable model for both basic research and preclinical testing of new therapeutic options for treatment of TCS patients.

## 1. Introduction

Despite being small anatomical regions, the sinonasal cavities harbor a great number of histologically different cancers, including epithelial types such as squamous cell carcinoma (SNSCC), salivary gland tumors and intestinal-type adenocarcinoma (ITAC); neuroendocrine types such as olfactory neuroblastoma (ONB), neuroendocrine carcinoma (SNEC) and malignant mucosal melanoma (MMM); and various forms of sarcoma [1,2]. Sinonasal tumors also frequently display more than one histological appearance, e.g., ITAC with ONB, SNEC with ONB, and SNEC with SNSCC or ITAC [3,4,5,6,7,8,9,10]. Such mixed or collision tumors occur at the time of presentation, but cases where locoregional recurrences arise histologically different from their corresponding primary tumors have also been described, for example, ITAC recurring as undifferentiated carcinoma (SNUC) or ONB recurring as non-ITAC adenocarcinoma [11,12].

Teratocarcinosarcoma (TCS) is arguably the sinonasal tumor with the most variable histology and is defined as having an admixture of epithelial, mesenchymal, neuroendocrine and germ cell elements. It can feature glandular and squamous, fibroblastic and myofibroblastic, as well as immature round neuroepithelial and sarcomatous areas [1,2]. Even among sinonasal tumors, which are infrequent, TCS is particularly rare, with a total of 127 patients described by Chapurin et al. in 2021 [13]. In the clinic, TCS is predominantly observed in men with an average age of 50–55 years [13,14]. Most frequent symptoms of SNTCS are nasal cavities obstruction, epistaxis (fever) and headache. Treatment typically consists of surgery followed by radiotherapy, but proton beam therapy and chemotherapy are also applied [13,14,15]. Multimodal treatment appears to have the best results; however, two-year survival is still as low as 55% [13,14,16,17].

New options for neoadjuvant, concomitant or adjuvant therapy could be the application of modern inhibitors of molecular targets. The first case report studies have indicated activating mutations in *CTNNB1* encoding β-catenin and *PIK3CA*; however, it is not yet clear how recurrent they are in TCS [18,19,20,21,22,23]. In 2020, a very high frequency (82%) of loss of SMARCA4 expression was reported and, in 2023, this study was followed by a genetic analysis demonstrating *SMARCA4* as well as *CTNNB1/APC* mutations in 65% and 41% of tumors, respectively [24,25]. *SMARCA4* is a component of the chromatin remodeling SWI/SNF complex, which governs cellular lineage differentiation and proliferation [26,27]. Alterations in two other factors in this complex, *SMARCB1* and *ARID1A*, have also been described in TCS [25]. *CTNNB1*, encoding β-catenin, and *APC* are members of the Wnt pathway, and mutations in these genes have been described previously in other epithelial sinonasal tumors, including ITAC and the proposed new tumor entity olfactory carcinoma [28,29,30,31,32,33]. Both SWI/SNF and Wnt pathways are involved in the regulation of transcription, among other functions. No FDA- or EMA-approved therapies have yet been developed targeting these pathways, but several therapeutic possibilities are being investigated in preclinical and clinical studies [27,34,35,36].

In vitro and in vivo models are essential for preclinical anticancer drug testing and also functional studies on the signaling pathways involved in processes such as cell differentiation and tumor invasion. Here, we present a new tumor cell line named TCS627 derived from a previously untreated primary TCS originating in the nasal cavity. We show how histological and genetic characteristics compare to its original primary tumor. From the many somatic mutations identified by whole-exome analysis, we focused on two inactivating mutations affecting *SMARCA4* and *CDKN2A* to investigate the efficacy of EZH1/2 and CDK4/6 inhibitors as candidate therapeutic options for the treatment of TCS patients.

## 2. Material and Methods

### 2.1. Clinical Description

A primary tumor sample and peripheral blood were obtained from a 68 year old male patient who presented with nose bleeding and fever episodes. He was a habitual smoker and occasional drinker, but had no history of exposure to wood, leather or textile dust, or industrial chemical substances as glues, formaldehyde, chrome or nickel. He had been previously treated with xazal and rinobanedif without any benefit. Rhinoscopy showed an irregular large 7 × 4 × 2 cm bleeding tumor occupying the whole left nasal cavity invading 2 × 2 cm into the brain through the cribriform plate (Figure 1). There were no lymph node or distant metastases nor changes in the oropharynx. The tumor was surgically resected but no free margins could be warranted. Adjuvant radiotherapy ended six months after surgery, at which point an MRI scan revealed a large intracranial progression and local recurrence. The patient died a short time later. This study was performed in accordance with the approved guidelines of the Ethics Committee of the Hospital Universitario Central de Asturias, and informed consent was obtained from the patient.

### 2.2. Establishment of Cell Line TCS627

A fresh tumor sample from the operating theatre was cut into several small fragments, transferred to dry 25 cm^2^ culture flasks, covered with culture medium and incubated in 5% CO_2_ at 37 °C. Initial outgrowth of both tumor and fibroblast cells was observed after 7 days. Fibroblasts were removed by repeated selective trypsinization. At the moment of writing this manuscript, the cell line has been in culture for more than 60 passages without changing its growth rate or phenotypic characteristics. Possible mycoplasma contamination was regularly checked using the LONZA MycoAlert Mycoplasma Detection Kit (LONZA, Rockland, CE, USA) and always came out negative. The culture medium consisted of DMEM/F12 supplemented with Glutamax (Gibco/Fisher Scientific S.L., Madrid, Spain), 5% FBS, 0.4 μg/mL hydrocortisone, 5 μg/mL insulin, 8.4 ng/mL cholera toxin (Sigma-Aldrich; Darmstadt, Hesse, Germany), 24 μg/mL adenine (Santa Cruz Biotechnology, Inc., Heidelberg, Germany), 10 ng/mL EGF (Fisher Scientific S.L., Madrid, Spain) and 10 μmol/mL ROCK inhibitor Y-27632 (MedChemExpress/DISMED S.A., Gijon, Spain).

### 2.3. DNA Extraction and Cell Line Authentication

DNA was extracted with the High Pure PCR Template Preparation kit (Roche Diagnostics GmbH, Manheim, Alemania) from the cell line (passage 35), the primary tumor and from normal blood lymphocytes of the same patient. Short tandem repeat (STR) genotyping was performed using the Promega Powerplex 16 system (Promega Biotech Ibérica SL, Barcelona, Spain), analyzing fifteen STR loci (Penta E, D18S51, D21S11, TH01, D3S1358, FGA, TPOX, D8S1179, vWA, Penta D, CSF1PO, D16S539, D7S820, D13S317 and D5S818) and Amelogenin by PCR.

### 2.4. Genetic Characterization

Metaphase preparations were made according to standard procedures and conventional karyotyping using DAPI banding. Images were captured using the Olympus BX-61 fluorescence microscope. Whole-exome sequencing (WES) was performed using the SureSelect Human All Exon V6 Kit for Illumina Multiplexed Sequencing (Agilent Technologies, Santa Clara, CA, USA) following the manufacturer’s instructions (Protocol Version D0, November 2015), resulting in an average coverage of 150×. Bioinformatic analysis was carried out with the HD Genome One v4.x.y software certified with IVD/CE-marking (DREAMgenics, Oviedo, Spain), including quality control, alignment and somatic variant calling. We discarded sequence variants with an allele frequency of higher than 1% in the normal population as well as sequence variants with less than 5 reads or less than 10% of the total reads in the tumor sample. Finally, we considered only non-synonymous somatic variants appearing in the tumor but not in the normal germline sample of the patient. The detection of copy number alterations was performed using the Nexus Copy Number Discovery BDI8840-AS2 software (Bionano, San Diego, CA, USA) using the raw FastQ WES data as input. At least ten probes per segment were considered as the minimum number to define a copy number alteration. Gains were called if the log2 ratio was >0.2 and losses ≤ 0.2. High copy number gains were scored when the log2 ratio > 1.2 and homozygous deletions ≤ 1.2.

Two somatic mutations were confirmed by PCR using primers *CDKN2A* Exon 2: Forward 5′-ACCATTCTGTTCTCTCTGGCA-3′ and Reverse 5′-GATGGCCCAGCTCCTCAG-3′, and *SMARCA4* Exon 8: Forward 5′-GCTAGACGTCCCCTGCAC-3′ and Reverse 5′-TAGGCCTTAGCATTGAGGGC-3′. Amplification was carried out on a Simpliamp Thermal Cycler VXA24811 (Applied Biosystems/Fisher Scientific S.L. Madrid, Spain). The conditions were as follows: (95 °C for 5 min + (95 °C for 15 s, 60 °C for 1 min, 72 °C for 1 min) × 40 cycles + 72 °C for 7 min and finally 4 °C). The PCR products were purified with Exo-BAP Mix (EURx Ltd., Gdansk, Poland) and sequenced with the ABI PRISM 3100 and 3730 Genetic Analyzers (Applied Biosystems/Fisher Scientific S.L., Madrid, Spain). Sense and antisense sequencing were performed for confirmation. Human papillomavirus (HPV) DNA detection was checked by PCR amplification of b-globin forward primer PC04 5′- CAACTTCATCCACGTTCACC-3′ and reverse primer GH20 5′-GAAGAGCCAAGGACAGGTAC-3′). PCR with MY11/GP6+ primers (site directed L1 fragment of HPV) was performed to detect a broad spectrum of HPV genotypes. The conditions were as follows: (94 °C for 5 min + (94 °C for 30 s, 55 °C for 30 s, 72 °C for 1 min) × 40 cycles + 72 °C for 10 min + finally 4 °C). The amplified DNA fragments of approximately 200 bp were identified by electrophoresis in 1.5% agarose gel with UV.

### 2.5. Immunohistochemistry

Whole tissue sections 3 µm in size were cut from a formaldehyde-fixed paraffin-embedded (FFPE) tissue block of the primary tumor. In addition, 3 µm sections were taken from a second FFPE block prepared from cultured TCS627 cells included in histogel (Fisher Scientific S.L., Madrid, Spain). Immunohistochemistry (IHC) was performed on an automatic staining workstation (Dako Autostainer Plus; DakoCytomation, Glostrup, Denmark) with antigen retrieval using EnVision FLEX + Mouse (DakoCytomation, Glostrup, Denmark) for 20 min. The following antibodies were used: EMA clone E29, S-100 polyclonal GA504, Neuron-specific enolase clone BBS/NC/VI-H14, Vimentin clone V9, CD99 clone 12E7, Desmin clone D33, Myogenin clone F5D, alpha-fetoprotein (AFP) polyclonal GA500, Chorionic Gonadotropin (hCG) polyclonal GA508, CK8 clone DC10, Caldesmon clone h-CD, CK20 clone Ks20.8, CDX2 clone DAK-CDX2, synaptophysin clone SY38, chromogranin A clone DAK-A3, p63 clone DAK-p63, p53 clone DO-7 and Ki-67 clone MIB-1 (DAKO, Glostrup, Denmark); p40 clone BC-28, p16 clone E6H4 and SALL4 clone 6E3 (Roche, Mannheim, Germany); β-catenin clone β-catenin-1 (BD Biosciences, San Jose, CA, USA); SMARCA4 clone ab70558 (Abcam, Cambridge, UK); SMARCB1 clone D8M1X (Cell Signaling Technology, Cambridge, UK); and SMARCA2 polyclonal HPA029981 (Sigma-Aldrich; Darmstadt, Hesse, Germany). The stainings were evaluated in a double-blind manner by three observers (BV, VBL and SLLG), and discrepancies between the observers were resolved by a consensus review after simultaneous reevaluation.

### 2.6. Cell Proliferation and Drug Sensitivity Assay

The growth rate was assessed by seeding 150,000 cells in 12-well plates and cell count measurements at 24 h intervals for 4 days using an automated cell counter (Countess 3, Thermo Fisher Scientific Inc., Waltham, MA, USA). Population doubling time was determined by calculating and considering the exponential growth phase.

Assessment of growth inhibition by valemetostat and palbociclib (MedChemExpress/DISMED S.A., Gijon, Spain) was performed by seeding 30,000 cells/well in 8-well chambers (Ibidi GmbH, Gräfelfing, Germany) and, after adhering for 24 h, exposure to concentrations of 0.02, 0.2 and 2 μM for 48 h. Cells treated with 0.01% DMSO were used as controls. Cell proliferation was evaluated by EdU (5-ethynyl-2′-deoxyuridine) incorporation using Click-iT EdU Cell Proliferation Assay Kit (Invitrogen, Carlsbad, CA, USA) according to the manufacturer’s protocol. Briefly, cells were incubated with 10 μM EdU 40 min prior to fixation with 4% paraformaldehyde. The cells were permeabilized with 0.1% Triton X-100 in phosphate buffer, followed by EdU detection via a copper-catalyzed reaction and nuclei staining by DAPI (Thermo Fisher Scientific Inc., Waltham, MA, USA). Images were captured from 5–10 fields and further analyzed with Image J/Fiji 1.5.3 software. The percentage of EdU-labeled—and thus indicating DNA-synthesizing—cells was evaluated as the percentage of green, fluorescent nuclei over the total number of cells reflected by DAPI-stained nuclei. One-way ANOVA test, followed by Dunnett’s multiple comparisons test were used to determine significance between the percentages of EdU-positive cells in each treatment concentration relative to untreated control cells.

## 3. Results

### 3.1. TCS627 Cell Morphology, Differentiation and Proliferation Rate

The resected primary tumor showed signs of ulceration and histologically constituted epithelial, mesenchymal and neuroepithelial elements. Glandular areas with atypical epithelium and squamous differentiation were observed, as well as the presence of fusiform cells with fibroblastic and myofibroblastic traits (Figure 2). The most prominent component was neuroepithelial, with immature cells growing in rosette formation, showing an elevated proliferation with Ki67 of approximately 50%. Immunohistochemical analysis (Figure 3) revealed general positivity for epithelial membrane antigen (EMA) and CK8 with few cells also staining p40 and CK20, but absence of CK34BE12, CK19 and CDX2. Vimentin was strongly positive, S-100 and caldesmon were focally positive and desmin and myogenin were negative. Neuron-specific enolase (NSE) and CD99 stained 35% and 80% of cells, respectively, but synaptophysin and cromogranin were absent. Finally, SALL-4, glial fibrillary acidic protein (GFAP) and Glypican were focally positive and human chorionic gonadotrophin (HCG), alpha-feto protein (AFP) and placental alkaline phosphatase (PALP) were negative. A final diagnosis of TCS was reached.

Cell line TCS627 also showed several different cell morphologies, including tightly packed epithelial-like cells; fusiform cells; and very small, contrast-rich cells with little cytoplasm. The relative proportions of these cell types changed with the level of confluence in the culture flasks (Figure 4). TCS627 grew with an approximate population doubling time of 48 h. A pellet of cultured cells was fixed in formalin and embedded in a paraffin block for further immunohistochemical analysis. Immunohistochemical staining of diagnostic markers showed expression of EMA, vimentin, NSE and CD99, and absence of AFP, which is comparable to the results in the primary tumor. However, CK8, S-100 and SALL-4 that stained positive in the primary tumor were negative in TCS627 cells (Figure 5).

### 3.2. TCS627 Authentication

STR analysis performed on DNA from the cell line, primary tumor and blood lymphocytes from the patient confirmed that the cell line was indeed derived from the patient’s primary tumor. The 15 STR loci were 100% identical between DNA from normal blood lymphocytes and cell line TCS627. The primary tumor showed a discrepant allele (D3S1358) and one lost locus (D5S818) relative to the ones observed in normal blood lymphocyte DNA, which supposes a 93% concordance (Appendix A).

### 3.3. Genetic Characterization

DAPI banding of cell line TCS627 showed a tetraploid karyotype with five copies of chromosome arm 1q and whole chromosome 8, and six copies of whole chromosome 10 and 12. There were three copies of 1p, while two of the four copies of 15q were in translocation attached to 1q, apparently fused at both centromeres. Analysis of DNA copy number changes was possible by using the whole-exome sequencing data and showed common but also unique alterations in the primary tumor and the derived TCS627 cell line. As shown in Figure 6, the primary tumor demonstrated losses of chromosomes 19 and X, and gains at chromosomal regions 1q, 7p, 8 and 12, whereas cell line TCS627 harbored losses at chromosomal regions 1p, 9, 19, 22 and X, and gains at 1q, 10, 12, 16 and 21.

WES analysis of cell line TCS627, its corresponding primary tumor and normal DNA derived from blood lymphocytes yielded an average coverage of 146–164 reads. A total of 99 somatic mutations (showing no variant reads in normal DNA) included seventy-one non-synonymous, thirteen splicing, eight frameshift, five stop gained, one first MET and one in-frame delins variants. Table 1 presents eight mutations in cancer-related genes, including frameshift mutations in *ARID2* and *CDKN2A*; splice mutations in *SATB2* and *SMARCA4*; and missense mutations in *NOTCH3*, *STAG2*, *TET2* and *WNT7A*, all of which are present both in the primary tumor and in the cell line. A complete description of all WES results is presented in Appendix A. PCR Sanger sequencing confirmed the inactivating mutations in *SMARCA4* and *CDKN2A* (Figure 7), whereas no presence of HPV DNA was detected.

Immunohistochemical analysis showed diffused SMARCA4 expression in the primary tumor but almost complete loss of expression in TCS627 cells, while p16 showed nuclear positivity both in the primary tumor and TCS627 cells. Additional immunostainings on the primary tumor showed partial loss of SMARCB1 and SMARCA2, and patches of nuclear β-catenin positivity. In cell line TCS627, no SMARCB1 or nuclear β-catenin was detected, while SMARCA2 was almost completely lost. Finally, p53 expression was seen in a small proportion of cells both in the primary tumor as well as in cell line TCS627 (Figure 8).

### 3.4. TCS627 Growth Inhibition Assays

Based on the identified inactivating mutation in *SMARCA4*, the growth inhibitory potential of dual EZH1/2 inhibitor valemetostat and CDK4/6 inhibitor palbociclib was evaluated at 48 h exposure to concentrations of 0.02 µM, 0.2 µM and 2 µM. TCS627 cells were relatively resistant to valemetostat, with only 25% growth reduction at 2 µM concentration. In contrast, palbociclib showed a clear dose-dependent response. At the highest concentration, cell proliferation appeared to have come to a complete stop (Figure 9). 

## 4. Discussion

TCS is a rare and aggressive tumor type almost exclusively originating in the sinonasal cavities [1,2]. Facing a very poor prognosis, patients with TCS are in need of new options for treatment. Recently identified recurrent mutations in SWI/SNF and Wnt pathway related genes [18,19,20,22,24,25] may serve as targets for modern therapies with specific inhibitors, but preclinical development and testing of new compounds is hampered by the lack of appropriate experimental models. This is particularly the case for rare malignancies such as sinonasal tumors, of which only few cell lines have been established [37]. To our knowledge, TCS627 is the first in vitro model for sinonasal TCS to be described in the scientific literature.

First, we used STR analysis to confirm that the cell line was indeed derived from the original primary tumor. Next, we applied a panel of diagnostic immunohistochemical antibodies to show that the primary tumor expressed epithelial, neuroepithelial, sarcomatous and teratomatous markers, characteristic of TCS. The dominant component was neuroepithelial, which had a Ki67 index of approximately 50%. We expected the cell line to maybe represent one of the histologically different TCS components, but this was not the case; TCS627 cells look epithelial and fusiform, and there are tightly packed small cells that we believe represent neuroepithelial differentiation. With every passage of the culture, the newly seeded cells first appear mostly epithelial and, after some days, the fusiform and small cells begin to dominate and seem to form nests (Figure 4). Immunohistochemical expression of diagnostic markers EMA, vimentin, NSE and CD99, and AFP was comparable to the expression observed in the primary tumor (Figure 5). However, the moderate staining of CK8 and SALL-4, and the focal staining of S-100, p40 and CK20 in the primary tumor, were completely absent in the cell line. Also on the genetic level was TCS627, representative of its original primary tumor. WES analysis showed that 95 of the 99 somatic mutations were identical between the cell line and the primary tumor, albeit with higher allele frequencies in the cell line. Four gene mutations were present in the primary tumor only (Appendix A). At the level of chromosomal copy number gains and losses, however, cell line and primary tumor did show differences. Losses at 1p, 9 and 22 and gains at 10, 16 and 21 were present only in the cell line, whereas only the primary tumor showed gains at 7p and 8. Shared copy number changes were losses of chromosomes 19 and X, and gains of 1q and 12 (Figure 6). Previous cytogenetic analyses of TCS have indicated loss of chromosome arm 1p and gain of chromosome 12, and these two events are indeed present in cell line TCS627 [38,39]. Taking these results together, we conclude that TCS627 is a cell line that reflects most features of its original primary tumor and is representative of TCS in general.

WES and PCR sequencing indicated a splice mutation in *SMARCA4*, the most frequently mutated gene in TCS [25]. The *SMARCA4* gene is located on chromosome band 19p13.2. Copy number analysis showed a deletion of whole chromosome 19; however, the allele frequency of 0.65 in TCS627 cells indicated that the *SMARCA4* mutation was heterozygous (Table 1). In spite of the inactivating mutation, immunohistochemistry showed diffuse SMARCA4 positivity (although with variable intensities) in virtually 100% of cells of the primary tumor (Figure 8). Rooper et al. also described a case with *SMARCA4* mutation and intact expression, which they suggested to be due to a copy number gain of the gene [25]. Although copy number analysis indicated a loss of chromosome 19, with a tetraploid karyotype this probably means three instead of four copies of the gene, which is indeed what we found with the cytogenetic analysis of TCS627 cells (Figure 6). Alterations in other components of this complex such as *SMARCB1* and *ARID1A* have also been reported in TCS [25]. Our present case carries a co-occurring inactivating *ARID2* and *SMARCA4* mutation. Concomitant mutations in two or more SWI/SNF genes have not been described in TCS to date, but do occur in SMARCA4-deficient carcinoma, olfactory carcinoma and neuroendocrine carcinomas [25,40,41,42,43,44,45]—tumors that with TCS may be regarded as different morphotypes of one tumor entity [24,25,46]. In addition to the *SMARCA4* mutation and retained SMARCA4 expression, the primary tumor in this study also displayed partial loss of SMARCB1 and SMARCA2 expression. Since we found no mutations in these two genes, it may be speculated that the partial loss of expression is caused by promoter hypermethylation, as has been reported previously in lung cancer [47]. Different from the original primary tumor, cell line TCS627 demonstrated loss of SMARCA4 and SMARCA2 in a very large majority of cells, while SMARCB1 expression was completely absent (Figure 8). Notably, chromosomal regions 19p, 9p and 22q, where *SMARCA4*, *SMARCA2* and *SMARCB1*, respectively, are localized, all showed copy number loss as deduced from the WES data, which reflect three chromosome copies in a tetraploid context by cytogenetic analysis (Figure 6).

The second most frequently mutated gene in TCS is *CTNNB1*, encoding β-catenin [18,19,20,22] and frequently co-occurring with *SMARCA4* mutations in TCS and in SMARCA4-deficient carcinoma, olfactory carcinoma and neuroendocrine carcinomas [25,40,42,43,44,45,48,49]. Our TCS case did not harbor this mutation and did not show nuclear β-catenin staining; however, there were small patches of nuclear β-catenin in the primary tumor. TCS cases with only some focal staining of nuclear β-catenin with or without *CTNNB1* mutation have been described in the literature, and this may be related to the interaction of *SMARCA4* with the Wnt pathway [19,21,25,50]. WES analysis of TCS627 did indicate a mutation in *WNT7A*, a factor that can activate the canonical and non-canonical Wnt pathways described as a tumor suppressor in lung cancer but also as an oncogene in ovarian, breast and brain tumors [51,52]. The effect of the non-synonymous *WNT7A* mutation on protein function in the TCS627 cell line will require further study.

TCS627 also harbored an inactivation frameshift mutation in *CDKN2A*, the gene encoding p16. To our knowledge, this gene has not yet been implicated in TCS, although Sun et al. presented a case with a mutation in *RB1*, which together with *CDKN2A* is involved in cell cycle regulation. Just as in TCS627, this mutation occurred simultaneously with *SMARCA4* [53]. Mutations in *CDKN2A* or *RB1* co-occurring with *SMARCA4* or *ARID1A* have been reported previously in olfactory neuroblastoma and neuroendocrine carcinoma [40,43]. The diffuse nuclear positivity in both the primary tumor and cell line TCS627 detected by immunohistochemical analysis (Figure 7), together with the observed absence of HPV by DNA-PCR analysis, probably reflects an inactive form of p16 due to the *CDKN2A* mutation. *TP53*, another cell cycle regulator, was not mutated and showed normal protein expression in TCS627 and the primary tumor.

Other cancer-related gene mutations identified by our WES analysis included *SATB2*, *NOTCH3*, *STAG2* and *TET2*. Mutations in *NOTCH3* and *TET2*, as well as related genes *NOTCH1* and *TET1*, have been identified previously in TCS and related sinonasal neuroendocrine tumors [25,40]. Both families of genes are known for their roles in cell differentiation and cancer. *SATB2* is a transcription factor regulating osteoblastic, craniofacial and nervous system cell differentiation [54]. Possibly through interaction with the Wnt pathway, *SATB2* is also involved in tumorigenesis of many adenocarcinomas [55]. In sinonasal tumors, it is used diagnostically to distinguish ITAC from non-ITAC adenocarcinomas [56]. Reduced expression of SATB2 has been associated with poor prognosis in colorectal adenocarcinoma, while in head and neck squamous carcinoma, the loss of SATB2 expression was associated with recurrence and high tumor grade [55]. Finally, *STAG2* is a component of the multiprotein cohesin, which is responsible for cohesion of sister chromatids and, thus, chromosome stability. It is also involved in gene expression regulation through the formation of DNA loops. *STAG2* is frequently inactivated in cancer, particularly glioblastoma, urothelial carcinoma, Ewing sarcoma and myeloid leukemia [57].

Gene mutations identified in the present study and in previous studies on TCS may represent targets for therapy with specific inhibitors. Having shown that cell line TCS627 displays the characteristic histological features and genetic abnormalities, and is thus a valid model for TCS, we proceeded to investigate the anti-proliferative efficacy of candidate inhibitors. Several treatments for SMARCA4-deficient tumors have been proposed in the literature. These include platinum-based chemotherapy, taxanes and PARP-inhibitors targeting DNA repair and damage response, CDK4/6 inhibitors that may further reduce an already low CDK4/6 activity resulting from SMARCA4-deficiency, and EZH1/2 inhibitors that suppress PRC2 activity on which SWI/SNF deregulated tumors are dependent [27,58,59]. As a first approach, we studied the EZH1/2 inhibitor valemetostat and CDK4/6 inhibitor palbociclib. Our assays showed very little growth inhibition upon 48 h exposure to valemetostat. Two studies using panels of SMARCA4- and/or SMARCA2- mutated cell lines derived from small cell carcinoma of the ovary-hypercalcemic type (SCCOHT), lung, gastric and bladder cancer have shown that the efficacy of the EZH inhibitor tazemetostat was far less when only one of the two components was affected [60,61], similar to cell line TCS627. Moreover, immunohistochemical analysis indicated that although TCS627 cells are largely SMARCA4 and SMARCA2 deficient, a small percentage of cells still retained expression. This may explain the modest response to valemetostat; however, more study is required. In contrast, 48 h exposure to 2.0 µM palbociclib completely stopped proliferation, and 0.2 and 0.02 µM concentrations reduced cell growth. This strong response may be related to the SMARCA4 deficiency of TCS627, similar to results reported in *SMARCA4*-mutated SCCOHT cell lines [62,63]. On the other hand, the inactivating *CDKN2A* mutation in TCS627, although heterozygous, may play a role. Further functional studies of the activities of both signaling pathways will be needed to clarify the observed effects of valemetostat and palbociclib.

## 5. Conclusions

This paper describes the first stable tumor cell line derived from sinonasal TCS. The cells express epithelial, neuroepithelial, sarcomatous and teratomatous markers and harbor genetic features that are characteristic of TCS, particularly *SMARCA4* mutation. We believe TCS627 is a valuable model for basic research into the SWI/SNF and other signaling pathways that play a role in TCS tumorigenesis and also for preclinical investigation of candidate inhibitors that may become new therapeutic options for treatment of TCS patients.

## Figures and Tables

**Figure 1 cells-13-00081-f001:**
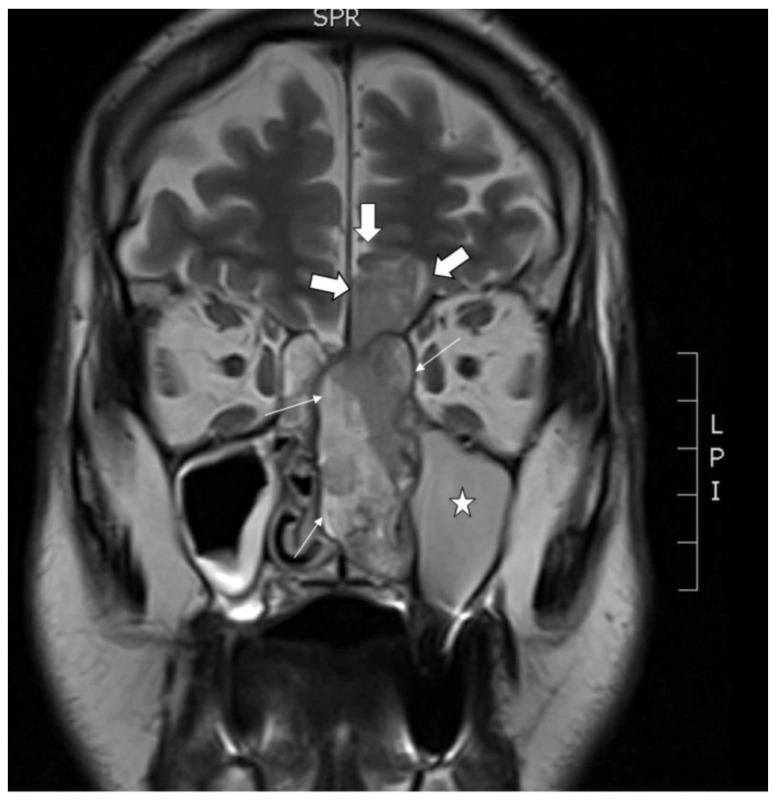
Coronal MRI showing the tumor in the left nasal cavity with a nasal component (thin arrows) and an intracranial component (thick arrows) in the shape of an hourglass. Asterisk in the maxillary sinus filled with mucus retention.

**Figure 2 cells-13-00081-f002:**
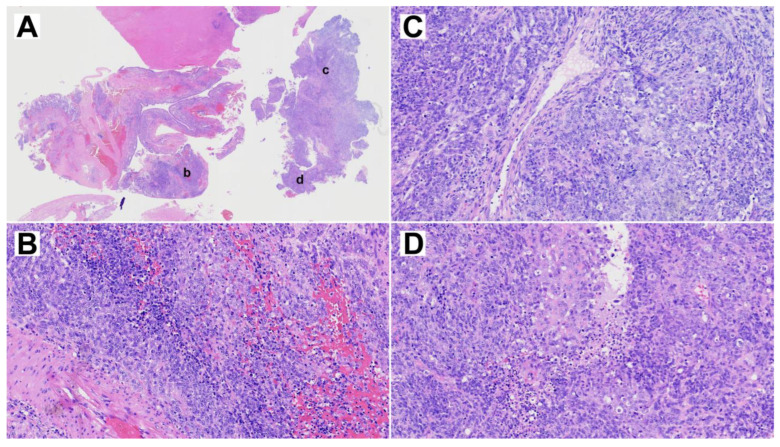
Representative H&E staining of the primary TCS. (**A**) Overview whole section at 1× magnification. (**B**–**D**) Three selected areas, indicated by lower case letters in panel A, at 10× magnification.

**Figure 3 cells-13-00081-f003:**
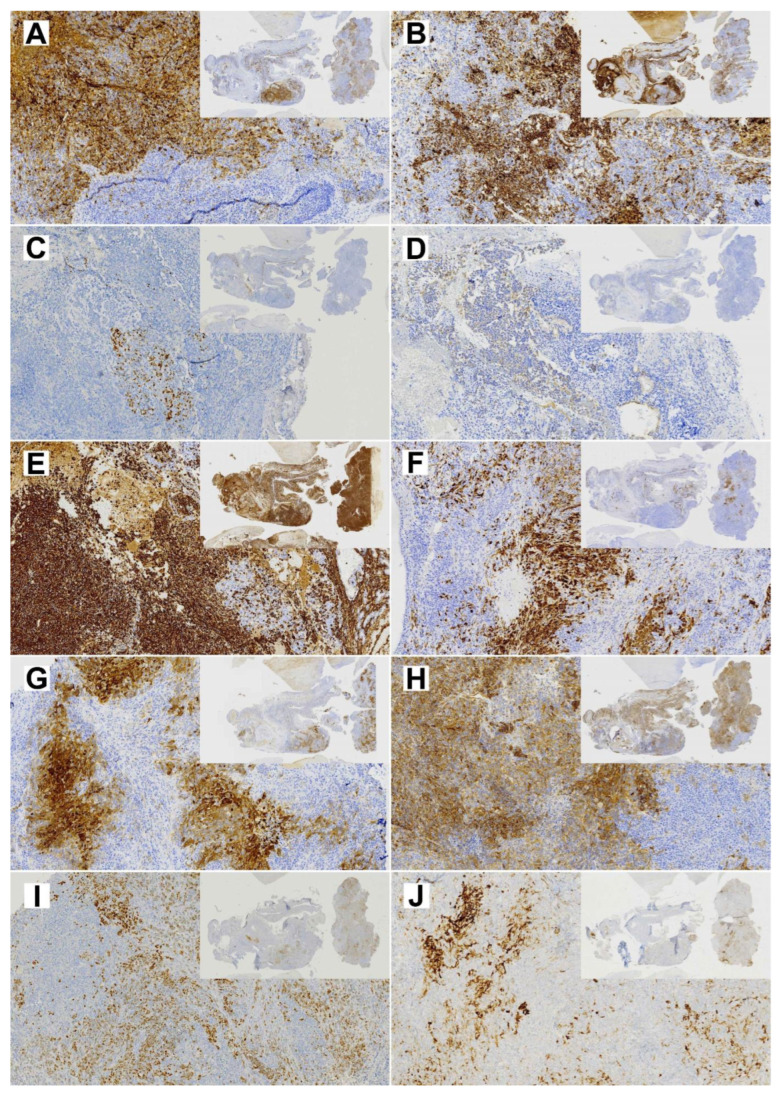
Diagnostic immunohistochemical stainings of the primary TCS (insert 1×, whole image 10× magnification). (**A**) EMA; (**B**) CK8; (**C**) p40; (**D**) CK20; (**E**) Vimentin; (**F**) S-100; (**G**) NSE; (**H**) CD99; (**I**) SALL-4; (**J**) Glypican.

**Figure 4 cells-13-00081-f004:**
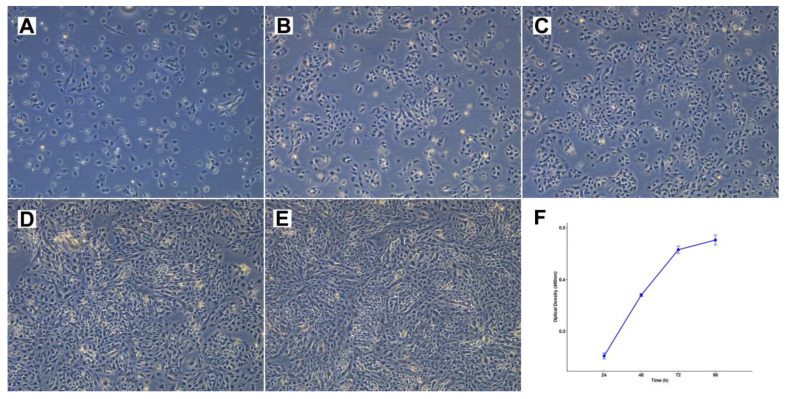
Cell morphology and growth rate of cell line TCS627. (**A**–**E**) Respectively, 1, 3, 4, 7 and 8 days after seeding at 1:6 dilution, 10× magnification. (**F**) Growth rate curve of TCS627 cells at passage 47; the population doubling time was approximately 48 h in the exponential growth phase.

**Figure 5 cells-13-00081-f005:**
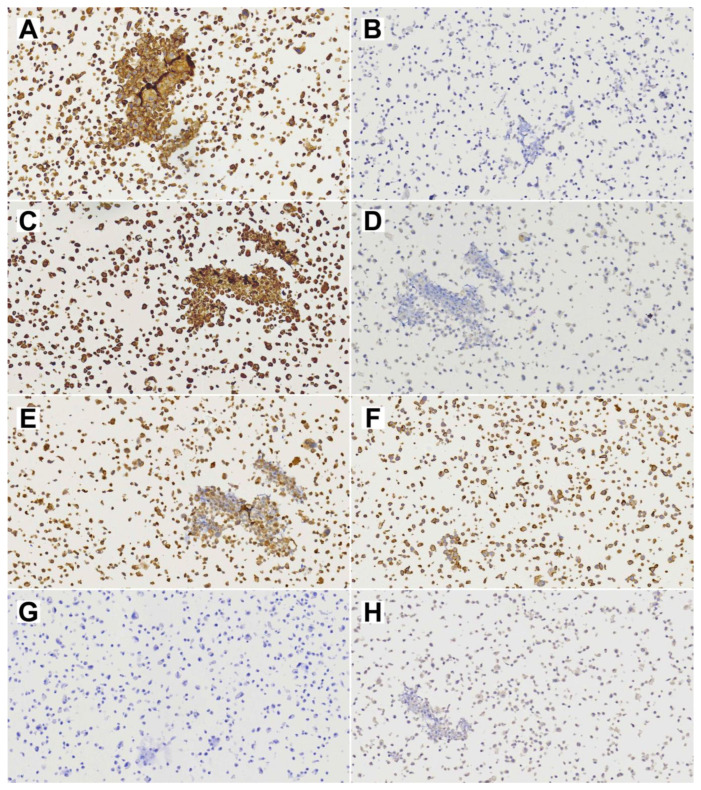
Diagnostic immunohistochemical stainings of cell line TCS627. (**A**) EMA; (**B**) CK8; (**C**) Vimentin; (**D**) S-100; (**E**) NSE; (**F**) CD99; (**G**) SALL-4; (**H**) Alpha-feto protein. All images at 20× magnification.

**Figure 6 cells-13-00081-f006:**
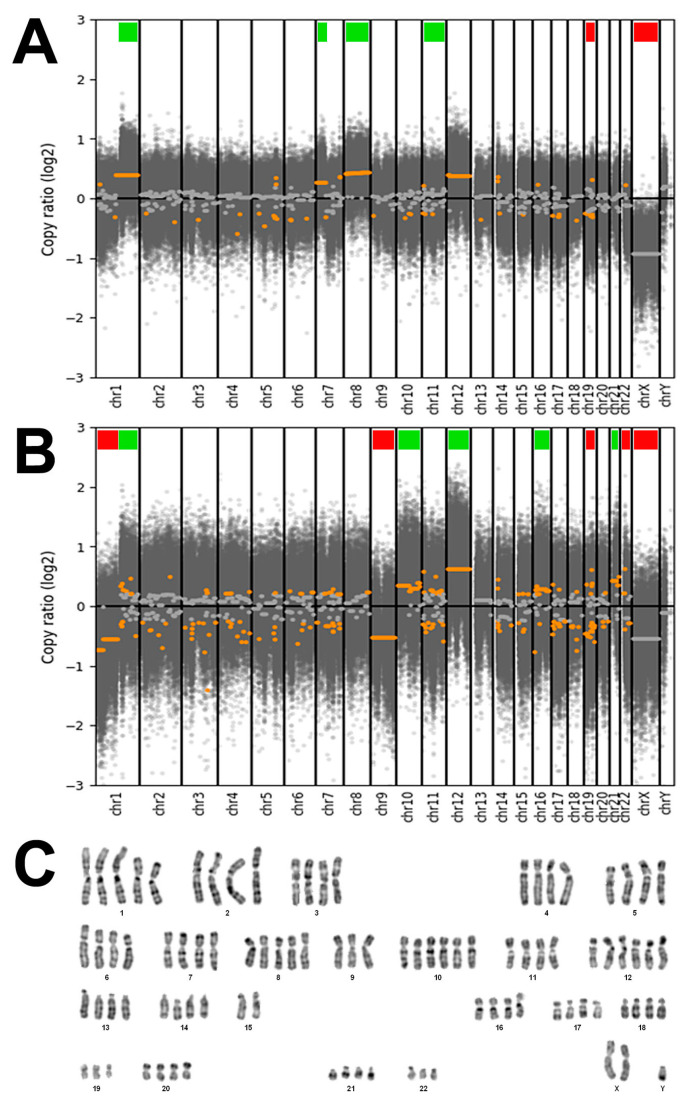
(**A**) DNA copy number changes of the primary tumor showing losses of chromosomes 19 and X (indicated by red bars), and gains at chromosomal regions 1q, 7p, 8 and 12 (indicated by green bars). (**B**) DNA copy number changes of cell line TCS627 showing losses at chromosomal regions 1p, 9, 19, 22 and X (indicated by red bars), and gains at 1q, 10, 12, 16 and 21 (indicated by green bars). (**C**) Representative DAPI banding of cell line TCS627 showing a tetraploid karyotype with five copies of chromosomes 1q and 8, and six copies of chromosomes 10 and 12. There are three copies of 1p, 19 and 22, while two of the four copies of 15q are in translocation attached to 1q (asterisks). Chromosome Y appeared in two copies in other karyotyped cells.

**Figure 7 cells-13-00081-f007:**
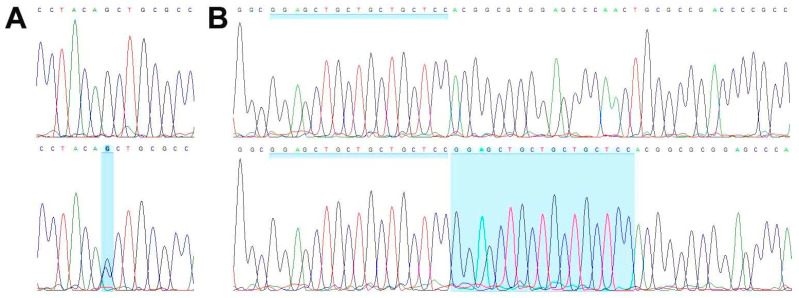
Wild-type sequence (**top row**) and inactivating mutations (**bottom row**) in *SMARCA4* (**A**) and *CDKN2A* (**B**).

**Figure 8 cells-13-00081-f008:**
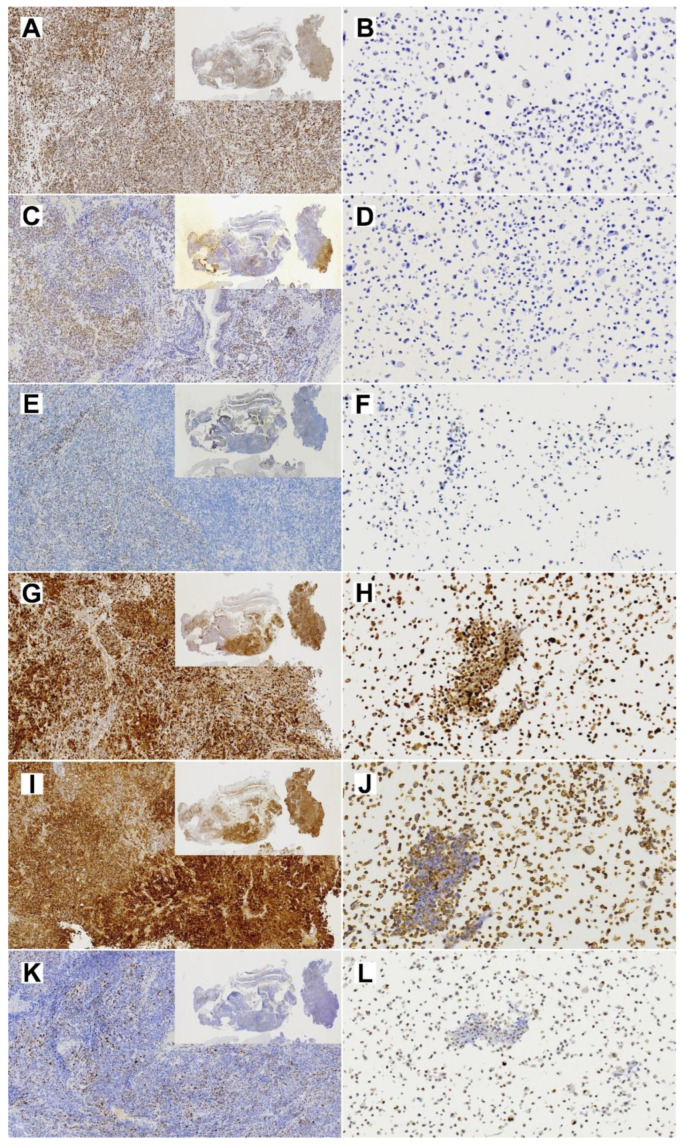
Immunohistochemical stainings of expression of cancer-related genes in the primary TCS tumor (**left column**; insert 1×, whole image 10× magnification) and the derived TC627 cell line (**right column**; 20× magnification). (**A**,**B**) SmarcA4; (**C**,**D**) SmarcB1; (**E**,**F**) SMARCA2; (**G**,**H**) p16; (**I**,**J**) Bcat; (**K**,**L**) p53.

**Figure 9 cells-13-00081-f009:**
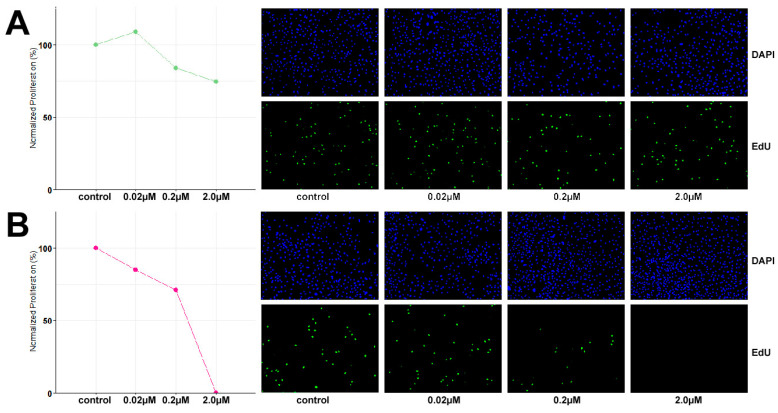
Dose response to dual EZH1/2 inhibitor valemetostat (**A**) and CDK4/6 inhibitor palbociclib (**B**).

**Table 1 cells-13-00081-t001:** Eight somatic mutations occurring in cancer-related genes identified by whole-exome sequencing of cell line TCS627, its corresponding primary tumor and normal DNA derived from blood lymphocytes.

Gene	c.Hgvs	p.Hgvs	Variant Reads Normal	Variant Reads Primary Tumor	Variant Reads TCS627	Variant Frequency Normal	Variant Frequency Primary Tumor	Variant Frequency TCS627
*ARID2*	c.1518_1519del	p.Gln507Alafs*13	0/133	38/161	73/184	0	0.24	0.4
*CDKN2A*	c.223_239dupGGAGCTGCTGCTGCTCC	p.Arg81Glufs*?	0/369	63/250	40/104	0	0.25	0.38
*NOTCH3*	c.1057G>A	p.Asp353Asn	0/343	112/310	154/229	0	0.36	0.67
*SATB2*	c.1741-12C>G		0/77	21/75	31/52	0	0.28	0.6
*SMARCA4*	c.1246-1G>C		0/222	68/203	133/205	0	0.33	0.65
*STAG2*	c.2537G>T	p.Gly846Val	0/57	39/59	53/53	0	0.66	1
*TET2*	c.863C>T	p.Pro288Leu	0/141	37/131	69/177	0	0.28	0.39
*WNT7A*	c.1043G>T	p.Cys348Phe	0/175	83/193	148/211	0	0.43	0.7

## Data Availability

The data presented in this study are available on request from the corresponding author.

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
