# Peer review of "Characterization of a Preclinical In Vitro Model Derived from a SMARCA4-Mutated Sinonasal Teratocarcinosarcoma"

_cells, 2023, doi:10.3390/cells13010081_

Round 1

Reviewer 1 Report

Comments and Suggestions for Authors

This is a study on the development and characterization of a sinonasal teratocarcinomasarcoma cell line. The paper is well-written, methods are robust, and the results are presented clearly and with sufficient details. This is a valuable addition to a relatively small number of available sinonasal cancer cell lines and represents a grear research model for studying the biology of this tumor type as well as it might help better understanding the SWI/SNF pathway in general.

There are a few minor issues to address:

1.     Introduction, page 2: 

a.     “prroton” has a typo error

b.     “olfactory carcinoma” has been only proposed as new entity by some head and neck pathologists but is still far from being defined. This may need to be re-phrased.

2.     Results, page 7.

Figure 2B, 2C, and 2D. these are all on 10X and look very similar at this magnification. Histologic features to demonstrate different components of teratocarcinomasarcoma might be presented at higher magnification (20X or 40X).

4.     Gene names should be in italic throughout the manuscript.

5.     References, pages 17-20. Numbers are duplicated.

Reviewer 2 Report

Comments and Suggestions for Authors

In this article, the authors describe the establishment and characterization of a new cell line derived from an ethmoidal teratocarcinosarcoma, and the in vitro sensitivity of tumor cells to valemetostat and palbociclib.

The manuscript is well written, and provides a precise description of the methods used to establish and characterize the cell line. The discussion is very comprehensive, addressing all the relevant issues.

The article could be accepted in its present form, but here are just a couple suggestions:

- in the introduction, the authors could point out that teratocarcinosarcomas are particularly rare, even among sinonasal cancers. Is it possible to give an estimate of their incidence?

- in the discussion, the authors (who have a unique experience in this field) could underline the fact that there are very few cell lines derived from sinonasal cancers, although this is a particularly interesting model in rare cancers.

Minor point: l57 “teratocarcinosarcoma”; and a curiosity: was the ROCK inhibitor necessary even after many passages, and do you think it could have an impact on the TCS tumor cell profile?
